# Impact Force Analysis in Inertia-Type Piezoelectric Motors

**Burhanettin Koc *** and **Bülent Delibas**

Physik Instrumente GmbH & Co. KG, 76228 Karlsruhe, Germany
* Correspondence: b.koc@pi.ws

**Abstract:** In an inertia-type motor, a piezoelectric multilayer actuator is espoused to a transient vibration velocity as high as 1.0 m/s during slip time. This vibration velocity makes the inertia-type motors dynamic but not quasi-static. We propose a kinetic model to describe the condition under which slippage can occur between a slider and a stator. The transient current absorbed by the multilayer actuators in a stator during slip time defines the slippage behavior of the slider. A new thickness-mode force factor expression ($A_{33}$), which is a relation between the transient current and the transient vibration velocity, is described in electrical domain. Impact force acting on a friction coupler produced by the actuators in the stator is proportional to the rate of change in the transient current during the sliding time. Additionally, we present the structure and characteristics of a two-phase inertia-drive-type piezoelectric motor, on which the proposed model was evaluated. Driving the multilayer actuators with truncated and mirrored sawtooth signals enhances the system dynamics. As one actuator expands and the other shrinks, their respective hysteretic nonlinearities are canceled. The motor operating frequency can be as great as 30 kHz and typically load characteristics are unloaded velocity greater than 16.0 mm/s and generated force higher than 3.0 N.

**Keywords:** piezoelectric; inertia drive; motor; transient current; transient vibration velocity; slip time

## 1. Introduction

Piezoelectric motors use the inverse piezoelectric effect. Here, microscopically small periodical movements are converted into continuous or stepping rotary or linear motions through frictional coupling between a displacement generating (stator) element and a moving (slider) element.

The idea of converting electrical oscillations into mechanical movement has been known for almost a century [1], and many of the original piezoelectric motor structures appeared in the 1980s [2–4]. Based on their drive principles and the type of microscopic displacement at the stator-slider interface, these motors can be categorized into three groups: (1) piezo walk drives, (2) resonance drives and (3) inertia drives [5].

Inertia-drive-type motors, which have been studied since the second half of the 1980s, have the benefits of simple structure and drive electronics, such as one piezoelectric actuator with one driving source [6–9]. However, single-actuator structures suffer from direction-dependent step sizes, which lead to increased control algorithm complexity. Although rarely referred to, actuator hysteresis in a single-actuator inertia-drive-type motor can further increase system nonlinearity and limit fine positioning ability, thus increasing the complexity of control electronics [10].

Recently, to enhance the controllability of positioning with direction-dependent step sizes, many piezoelectric inertia-drive-type motor structures, where two piezoelectric elements are embedded in a stator [11–24], have been developed. To achieve the same motion in both directions, a sawtooth signal actuates the piezoelectric elements one at a time. Additionally, the use of flexure or leverage mechanisms forces these motors to be operated in the audible frequency range. Because the resonance modes for flexures have a range of several kilohertz, it is not possible to drive such structures at frequencies greater than 20 kHz with a sawtooth signal. Even if the applied electrical signal has a sawtooth

waveform, at contact points, the movement is converted into harmonic back-and-forth motion. This explains why many of these motors are operated with two sinusoidal signals, with a certain phase difference in between.

In this work, the structure of the two-phase inertia-drive-type motor consists of two commercially available multilayer actuators (PL033.31 PICMA® Chip, PI Ceramic, Lederhose, Germany) framed in a U-shaped base mass with a top beam exerting a pressing force (Figure 1) [25]. Because deformations on the multilayer actuators are converted into a rotational movement of the friction coupler directly without any flexure or leverage, the operating frequency of the motor can be as great as 30 kHz.

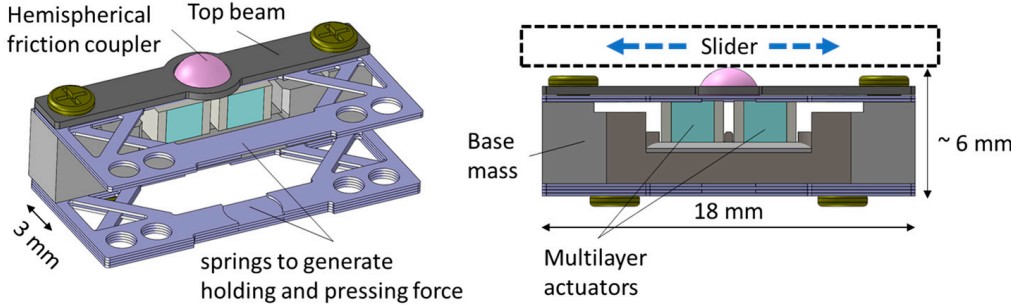

**Figure 1.** Perspective and side views of the stator showing the possible orientation of a slider. The stator is attached to a motor base from one side of the sheet metal flat springs.

Due to the fast response times of piezoelectric actuators, the motion transfer generated by the shrinkage and expansion of the actuators to the friction coupler is also fast. As is known from the operating principle of inertia-drive-type motors, the slider moves together with a friction coupler during stick-time. In contrast, during slip time, we expect the friction coupler to move fast enough so that the slider cannot follow it. However, the return motion during slip time is known. Due to the driving signals and design structure, the amount of slider return during the fast phase (slip time) could be minimized.

To the best of our knowledge, the nature of impact force generation is not known in the literature. Our analysis and the corresponding measurements reveal that the level of vibration to which a multilayer actuator is exposed during the slip time can exceed 1.0 m/s. The level of vibration velocity that a multilayer actuator is exposed to has not been addressed before.

The multilayer actuators in this motor are driven with truncated sawtooth signals of an anticyclic nature. Regardless of the direction of the slider, one actuator always generates a movement in a forward direction. During slip time, when one actuator expands quickly, the other actuator shrinks simultaneously. Similarly, during the stick time, the corresponding actuator expands or shrinks slowly, while the other does the opposite. The direction is changed by altering the expansion and contraction times. Due to the anticyclic nature of the driving signals, hysteretic movements of the multilayer piezoelectric actuators are compensated at the tip of the friction coupler. Further details of the structure and the operating principal are presented in the following section.

The remainder of the manuscript is organized as follows. A simplified kinetic model and the equivalent circuitry of a single multilayer actuator are presented in Section 3. We propose a new transient force factor, which gives the relation between the rate of change of the transient current and the transient force generated by the multilayer actuators during slip time. This proposed relation is verified experimentally by observing the transient voltage, current and vibration velocity on a multilayer actuator in Section 4. After giving the motor characteristics in Section 5, the manuscript is concluded with a discussion and features in Section 6.

## 2. Structure and Operational Principle

### 2.1. Stator Structure

As shown in Figure 1, the stator of the motor consists of a U-shaped base mass, two multilayer actuators of the same size ($3 \times 3 \times 2$ mm$^3$), sheet metal layers, a metal beam, and a ceramic hemisphere ($\phi$ 3 mm). The two actuators are placed adjacently in the U-shaped base mass, and the top of the actuators is covered with sheet metal layers and a metal beam, which apply a preloading force to the multilayer actuators in the direction of the length. The ceramic hemisphere is positioned in the middle of the beam and acts as the friction coupling element. The stator is mounted on the motor base using the sheet metal spring layers. The metal spring layers hold the stator parallel to the slider and press the stator body against the slider. This kind of integration simplifies the assembly process significantly, because the stator is mounted to the motor base by inserting and tightening two screws.

Driving the multilayer actuators using two truncated sawtooth signals of an anticyclic nature generates a microscopic rotational back-and-forth movement at the tip of the friction coupler. Frictional coupling then transfers this slow and fast back-and-forth movement to the slider.

Figure 2 shows the deformed states of the piezoelectric actuators in the stator. As piezoelectric actuators expand and shrink at the same time in the direction of their thickness, the thickness direction deformations are converted into a rotational movement on the hemispherical friction coupler. Another interesting point is that the center point of the hemispherical friction coupler is a hinge point.

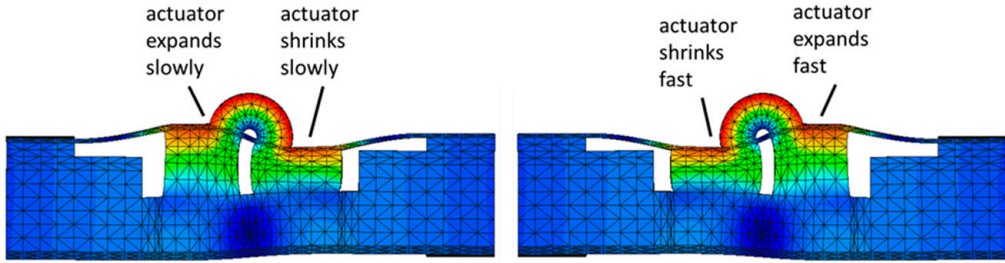

**Figure 2.** The deformed state of the actuators in the friction coupler resulted in friction coupler movement. Note that the intersection point of the actuators and the friction coupler is a hinge point. Dark blue color represents small displacements while red color represents larger displacement regions.

The U-shaped base mass has very small movements compared to the piezoelectric actuators and the friction coupler. Note that maximum displacement occurs on the surface of the hemispherical friction coupler.

After designing the stator structure, some dimensions, such as the height of the metal base and the gap distance between the actuators in the stator, were optimized with the help of ATILA-GID software (ATILA-GID 3.0.0, Micromechatronics Inc, State College, PA, USA). Using the harmonic analysis feature of the program, deformation of the friction coupler was calculated by varying the distance between the actuators in the stator. As seen in Figure 3, when the distance (u) is 0.5 mm, the normalized rotational movement of the friction coupler is approximately 25 nm under an electric field of 1.0 V/mm. If the gap distance is too wide, the force transferred from the actuators to the friction coupler will lessen. The small gap distance could be filled with epoxy, but this would likely change the actuator behavior. Therefore, a 0.5 mm gap was found to be ideal for both actuator performance and manufacturability.

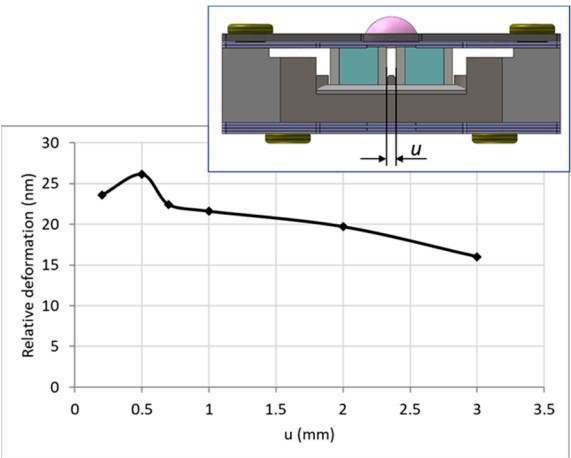

**Figure 3.** The relative deformation of the friction coupler is at a maximum when the distance between the actuators is 0.5 mm.

### 2.2. Motor Driving Principle

To ensure that both actuators expand and shrink synchronously, they are driven with two anticyclic truncated sawtooth signal waveforms. As seen in Figure 4, one period of the driving signal waveforms has three phases. During the first phase (p1), the signal magnitude on one actuator slowly increases from its natural level, while the signal magnitude on the other actuator decreases until the rated values are reached. When one actuator expands and the other shrinks, the friction coupler moves to one side. As the movement is slow, the slider also moves together with the friction coupler. In this phase, the slider makes a half step. This time is also called the first half of the stick time. In the second phase (p2), the magnitudes of the signals are quickly changing to their opposite sides. During this phase, the friction coupler position changes to the opposite side. If this time is short enough, the friction coupler will move to the opposite side so fast that the slider cannot follow and maintain its position. Regardless of the period of the driving signals, the time for phase 2, which is also called the slip time, ranges from 1 to 3 μs.

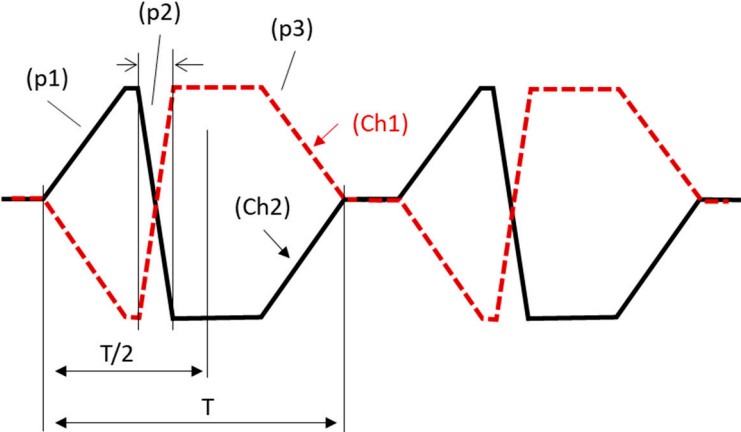

**Figure 4.** Motor driving signal waveforms of an "anticyclic" nature. Phase 2 (p2), which corresponds to the slip time, is not to scale. Before and after the stick and slip times, there are regions where the voltage levels are constant.

During the last phase (p3), the magnitudes of the driving signals are maintained for a short time, and they both slowly reach 0.0 V. In this phase, the friction coupler reaches its natural position. Since the movement is slow, the slider makes the second half of the step. This time is also known as the second half of the stick time. At the end of a period,

the voltage levels would stay at the reference level for a short time. Inserting a pause time would depends on the driving conditions.

It is well known that operating voltage range of multilayer actuators with positive potentials are larger compared to negative potential range. Our measurements were performed under the condition of +25 and −25 V. When an offset voltage is applied on both multilayer actuators in the stator, both multilayer actuators expand in their thickness directions. This expansion does not change natural position of the friction coupler in a stage. Then the maximum operating voltage range can be widened up to −20 to 100 V at an offset voltage of 40 V. Adding an offset voltage on both multilayer actuators does not generate a movement but increase pressing force, which would be good for motor performance.

Figure 5 shows the motor driving signal waveforms and the resulting slider position. The motor was driven with single-step burst waveforms. The frequency of the signals was 100 Hz, which corresponds to a period of 10,000 μs, and the three phases of the single step can be observed. The two anticyclic truncated sawtooth voltage driving signal waveforms, as seen in Figure 5, were first generated by a function generator (Tektronix AFG3000, Beaverton, OR, USA) and amplified by two power amplifiers (HAS 4014, NF Corp., Yokahoma, Japan). The position of the slider was measured together with the driving waveforms.

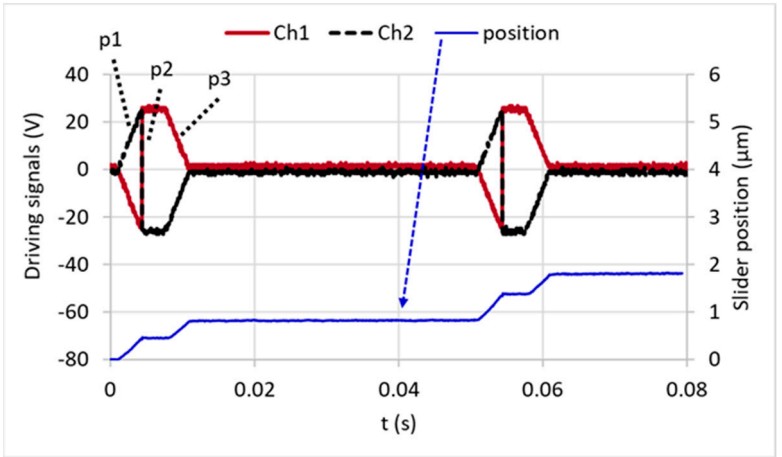

**Figure 5.** Single-step burst driving waveforms and the resulting slider position were measured for a slip time of 2 μs at 100 Hz.

p1: In this phase, the potential of one actuator increased to +25 V, and the other actuator's potential decreased to −25 V. Since one actuator shrunk and the other expanded, the friction coupler on the stator moved forward by approximately 0.4 μm. This amount is roughly half of the step size. During this period, the slider also moved the same amount.

p2: As soon as the actuator potentials reached their rated voltage values (+/−25 V), the magnitude of their voltage values interchanged very quickly. As a result, the friction coupler on the stator moved backward approximately 0.8 μm, leaving the slider in the forward position. The movement occurs so quickly that the slider cannot follow the friction coupler and maintain its position. In this phase, the slider might also make a small return if the slip time is not sufficiently short. During this measurement, the slip time was 2.0 μs, and a return motion was not observed.

p3: In the last phase, the potential values on the actuator stayed at their rated values for a short time, and both actuator voltages came to 0.0 V. During this time, the friction coupler moved to the zero position from −0.4 μm, and the slider made the second half of the step, which was again approximately 0.4 μm. If the following step would have been coming, the slider would have moved approximately 0.8 μm, while the first phase of the next step was generated, and the friction coupler would have moved to the +0.4 μm position.

### 3. Electromechanical Modeling and Verification of Friction Coupler Movement

*3.1. Kinetic Model*

This motor has two types of motion transfer. First, synchronous expansion and shrinkage of the multilayer actuators are converted into rotational back-and-forth movements of the friction coupler. Second, the rotational back-and-forth movements of the friction coupler are then transferred to a linear movement of the slider.

A simplified kinetic model of the motion transfer is formalized in Figure 6. The pressing force ($P$) against the slider causes a holding friction force $\left(F_{fs}\right)$ of the slider. The multilayer actuators are modeled with their mass ($m_p$), damping ($c_p$) and stiffness ($k_p$) constants. The stator base considered to be rigid, and the stiffness of the top beam is modeled as ($k_s$). Force ($F_p$) contributed by each actuator generates a moment to the friction coupler that is assumed to be rigid with its mass ($m_{fc}$).

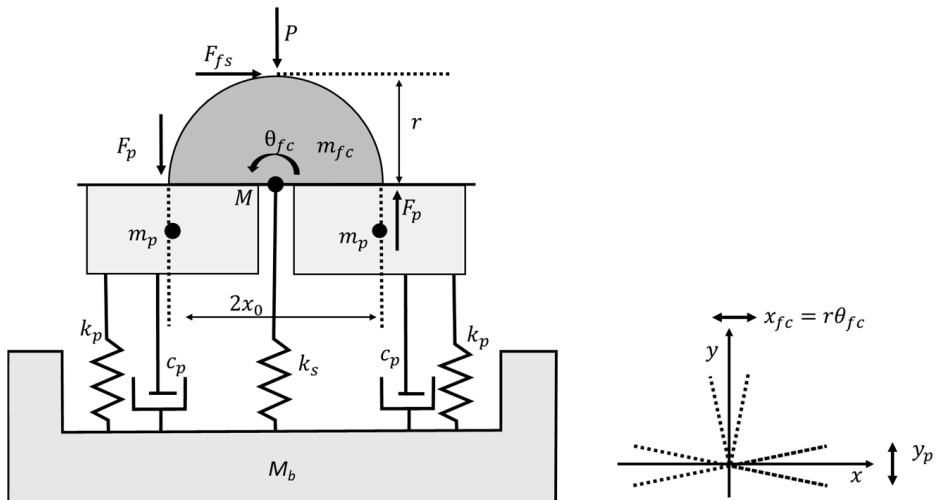

**Figure 6.** Kinetic model of the stator and the motion transfer from actuator shrinkage/expansion to angular deformation of the friction coupler. Distance between the center of gravity of the actuators is ($2x_0$). The actuator displacement ($y_p$) is converted into the rotational movement ($x_{fc}$) of the friction coupler.

The deformation in the thickness direction ($y_p$) of piezoelectric ceramics is converted into a rotational movement ($\theta_{fc}$) on the hemispherical friction coupler.

$$r\theta_{fc} = y_p \tag{1}$$

where, $r$ is the radius of the hemispherical friction coupler. If the actuator force acting on the slider is less than the friction force, there is no relative motion between the friction coupler and the slider.

The deformation in the thickness direction is converted into linear movement of the slider.

$$x_s = x_{fc} = r\theta_{fc} \tag{2}$$

where $x_s$ and $x_{fc}$ are the slider and friction coupler displacements, respectively. A movement is only possible if the tangential force, which acts on the friction coupler due to the movements of the piezoelectric actuators, is larger than the static friction force [9].

$$\left| r\ddot{\theta}_{fc} \right| > \frac{F_{fs}}{m_{sl}} \tag{3}$$

where $r\ddot{\theta}_{fc}$ is the tangential acceleration acting on the friction coupler. $F_{fs}$ and $m_{sl}$ are the static friction force and the slider mass, respectively. The equation of motion for the kinetic model can be expressed as

$$J_M \ddot{\theta}_{fc} + c_r \dot{\theta}_{fc} + k_r \theta_{fc} = 2F_p x_0 - rF_{fs} \tag{4}$$

where $J_M$ is the rotational moment of inertia of the actuators and friction coupler, $k_r$ is the rotational stiffness, and $c_r$ is the rotational damping factor coming from the elements of the stator structure, such as the epoxy layer and the sheet metal springs that exert holding and pressing forces. Since the equation for the kinetic model described in (4) is written for the complete slippage condition, the effect of the slider mass and friction nonlinearities are not included. In other words, if the impact force generated by the multilayer actuators is larger than the slider holding force then the friction coupler wins the breakaway force and leaves the slider behind.

When the dynamic force acting on the friction coupler is barely passing the holding force, a return movement of the slider takes place, in which case nonlinearities dominate the slider behavior. Friction nonlinearities in piezoelectric motors were addressed previously [26–28]. The focus in this section is to understand the impact force generation of the multilayer actuators during different slip times.

When a potential is applied to a multilayer actuator, according to the constitutive equations, the generated deformation ($y_p$) is proportional to the potential difference ($V_c$) between the actuator terminals:

$$y_p = n \, d_{33} V_c \tag{5}$$

where $n$ is the number of active layers in a multilayer piezoelectric actuator and $d_{33}$ is the thickness-mode piezoelectric constant (400 pC/N for PICMA$^{\circledR}$ Chip material, PI Ceramic, Lederhose, Germany). As the voltage on the multilayer actuator changes dynamically during the slip time, the deformation on the actuator should also be dynamic. Then, the generative force of a multilayer actuator can be obtained from the second derivative of the actuator deformation.

$$F_p = d_{33} m_p \left( \frac{d^2 V_c}{dt^2} \right) \tag{6}$$

where $m_p$ is the mass of the multilayer actuator, which consists of $n$ layers. Since potential on a multilayer actuator cannot maintain its linear increase because of internal resistance of a power source and cabling, a better representation of impact force generation can be obtained from the transient current. Then, the above relation can also be written from the rate of change of the transient current in a multilayer actuator.

$$F_p = \frac{d_{33} m_p}{C} \left( \frac{di_c}{dt} \right) \tag{7}$$

where $C$ is the capacitance of the multilayer actuator. As the current absorbed by a multilayer actuator is in phase with the vibration velocity, we can obtain the transient vibration velocity, $v_v$, as

$$v_v = \frac{n d_{33}}{C} (i_c) \tag{8}$$

and the thickness-mode force factor ($A_{33}$) can be defined in the electrical domain as follows.

$$A_{33} = \frac{C}{n d_{33}} \tag{9}$$

Although the newly introduced force factor expression in (9) is defined in electrical domain, the unit is identical to the thickness mode force factor expression known in the literature [29,30].

### 3.2. Equivalent Circuitry

Using the SPICE simulation software (LTSpice®, Analog Devices, Wilmington, MA, USA), voltage and current waveforms on one piezoelectric multilayer actuator were calculated by modeling the actuator as an RLC circuit, as shown in Figure 7. C (80 nF) is the capacitance of the multilayer actuator, and R (3.0 Ω) is the equivalent resistor representing the losses; 2.0 Ω is the actuator resistance obtained during the capacitance measurement and 1.0 Ω is the output resistance of the source voltage. In our case, the source voltage is the power amplifier internal resistance. The inductor (L = 3.0 μH) value in the circuitry is due to electrical wires, internal impedance of the power amplifier and the conductive electrode layers in the multilayer actuator. Since two multilayer actuators work in an anticyclic nature, the hysteresis term [31] in the equivalent circuitry was not included.

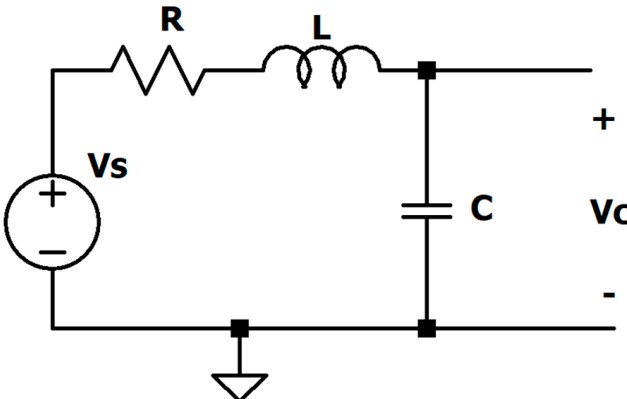

**Figure 7.** Simple RLC equivalent circuitry to model actuator behavior.

The slip times of the input voltages ($V_s$) were modeled as ramp signals with different slopes. The input signal can be represented as

$$V_s(t) = \left( \frac{t}{t_1} - \frac{1}{2} \right) V_0 [U(t) - U(t - t_1)] \tag{10}$$

where ($t_1$) is the slip time, ($V_0$) is the magnitude of the driving signal that is 50 V, and $U(t)$ is the unit step function. The effect of the signal waveform was only analyzed for 20 μs. After 20 μs, the signal magnitude slowly came to −25 V. Since the slider moves with the friction coupler, this time is not included in the model. From the equivalent circuitry, the transient current and voltage waveforms fulfill the two equations below:

$$i_c(t) = C \left( \frac{dV_c}{dt} \right) \tag{11}$$

and

$$LC \left( \frac{d^2 V_c}{dt^2} \right) + RC \left( \frac{dV_c}{dt} \right) + V_c = V_s(t) \tag{12}$$

Figure 8 shows the voltage signals with slip times of 1, 2, 3, 4 and 10 microseconds, which were modeled as ramp functions with initial slopes of 50, 25, 16.7, 12.5 and 5 V/μs. During slip times, the magnitudes of the voltages increase from −25 V to +25 V and remain at 25 V until 20 μs.

Figures 9 and 10 show the simulated transient voltage and current waveforms of the capacitance of the RLC circuitry. Note that $t = 0.0$ s was assumed to be the beginning of the slip time.

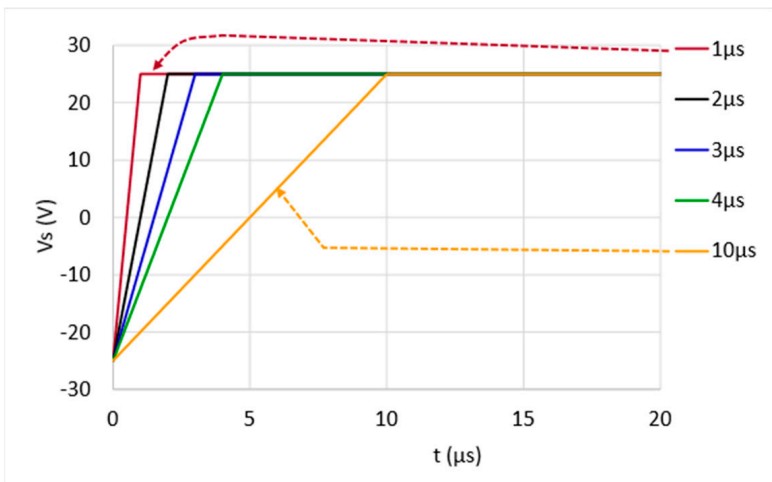

**Figure 8.** Ramp voltage signals with different initial slopes, while increasing from −25 V to +25 V. Magnitudes of the voltages after the slip time were constant at +25 V until 20 μs.

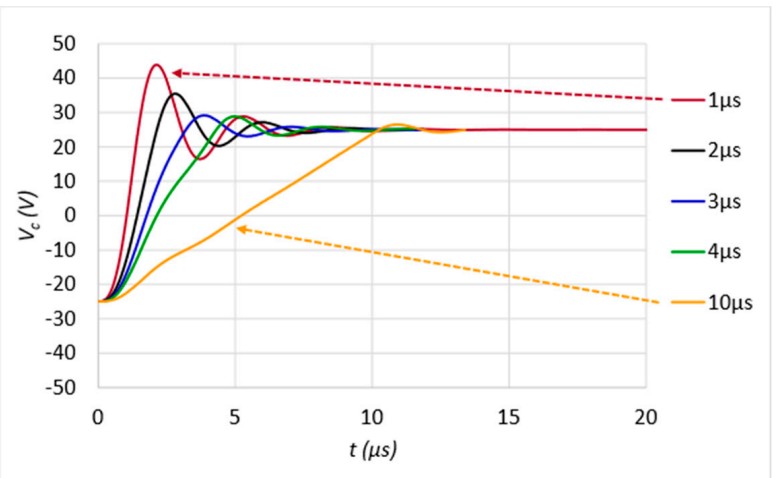

**Figure 9.** Simulated transient voltage waveforms on the capacitor element for slip times of 1, 2, 3, 4 and 10 μs.

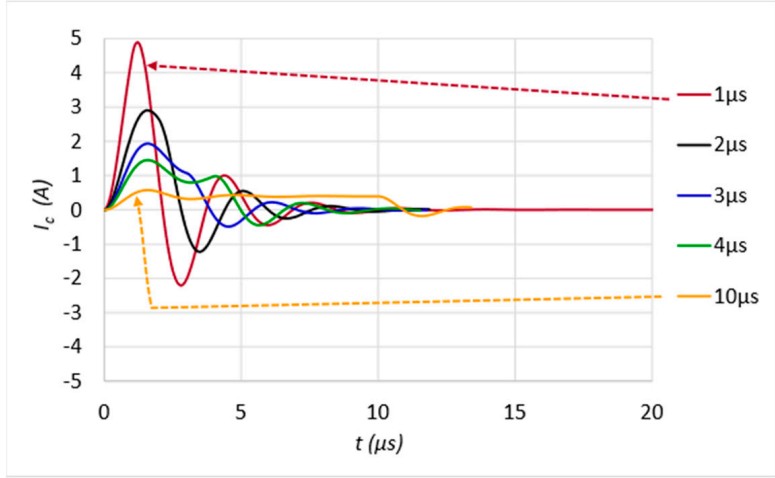

**Figure 10.** Simulated transient current waveforms of the capacitor element for slip times of 1, 2, 3, 4 and 10 μs. Signals with shorter slip times have a larger overshoot.

As the vibration velocity and absorbed current of a multilayer actuator are in phase, the transient vibration velocity can be obtained from (8), as shown in Figure 11. The overshoot value of the vibration velocity for a slip time of 1.0 µs can be higher than 1.0 m/s. To withstand such a high vibration level and to eliminate tensile stress in a multilayer actuator, it is important that the multilayer actuators used in an inertia-drive-type motor are prestressed. In the proposed design, the multilayer actuators are prestressed in two ways. One is by the top beam, and the other is by the springs exerting the holding and pressing forces.

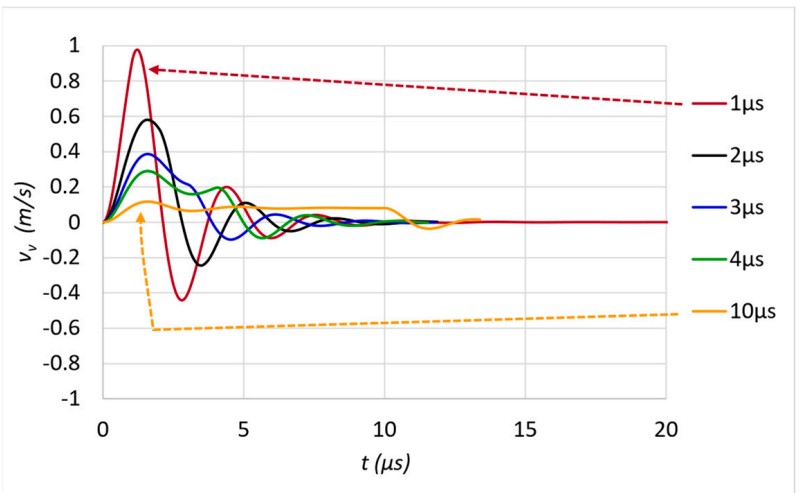

**Figure 11.** Estimated transient vibration velocities for different slip times calculated using the thickness mode force factor of the actual multilayer actuator and geometric parameter of the stator.

After taking the numerical derivative of the current waveforms for different slip times (shown in Figure 10), the transient force generated by the multilayer actuator can be estimated from (7). As shown in Figure 12, the shorter the slip time is, the higher the peak value of the generated impact force. As the generated force is the system's response, the time that the generated impact force reaches to its maximum value is always within the same response time. The response time at which the generated force reaches its maximum value is approximately 0.5 µs.

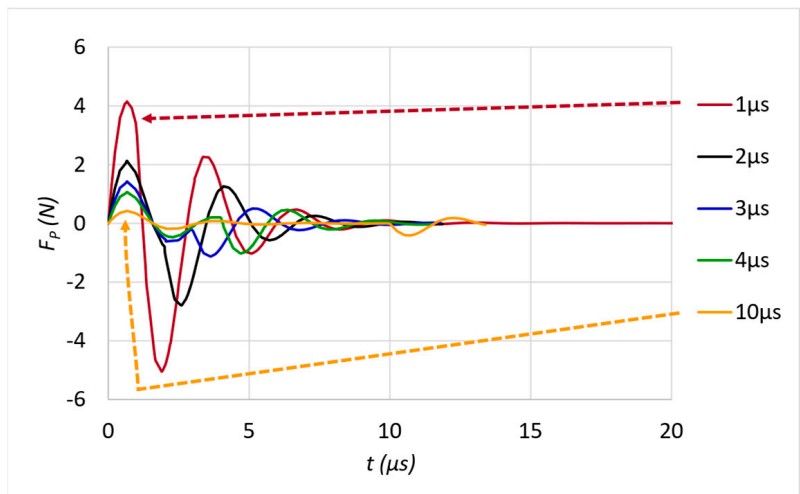

**Figure 12.** Estimated transient impact force, which is contributed by one multilayer actuator in the stator.

Regardless of the duration of the slip time, a slippage between the slider and the friction coupler can occur within 1 µs time. When the actuation force is reduced to less than the slider holding force, the slider sticks to the friction coupler and makes a return movement. As we will be discussing in Section 5, when slider is sticking to the friction coupler, response time of the slider return movement is much longer than the slip time.

We would emphasize that the force generated by the actuator during the stick time is always sufficient to move a slider. The transient impact force that we focused on here is the force generated during slip time. Since two multilayer actuators are used in the stator structure, the anticyclic driving nature combines the actuator forces. As a result, the force acting on the friction coupler is doubled. When the maximum generated impact force value is greater than the static friction force at the stator–slider interface, the friction coupler can leave the slider and make a microscopic return movement during slip time. If the peak value of the transient impact force is greater than the static friction force, which is also called the slider holding force, the friction coupler slips and leaves the slider behind.

## 4. Experiments

### 4.1. Transient Voltage, Current and Vibration Velocity Waveforms during Slip Times (On a Multilayer Actuator)

To compare the simplified equivalent circuitry mentioned above with the actual conditions, a multilayer actuator (PL033.31, PI Ceramic, Lederhose, Germany) used in the stator structure was characterized by attaching the actuator to a rigid mass followed by excitation with ramp signals, which had initial slopes of 50, 25, 16.7, 12.5 and 5 V/µs, as in the simulations. Using the measurement setups illustrated in Figure 13 (equipment used in the measurement setup is listed in the figure caption), transient current, vibration velocity and voltage waveforms were captured simultaneously by driving the multilayer actuator with a single step (Ch1 as shown in Figure 5), with different slip times excitations at 100 Hz.

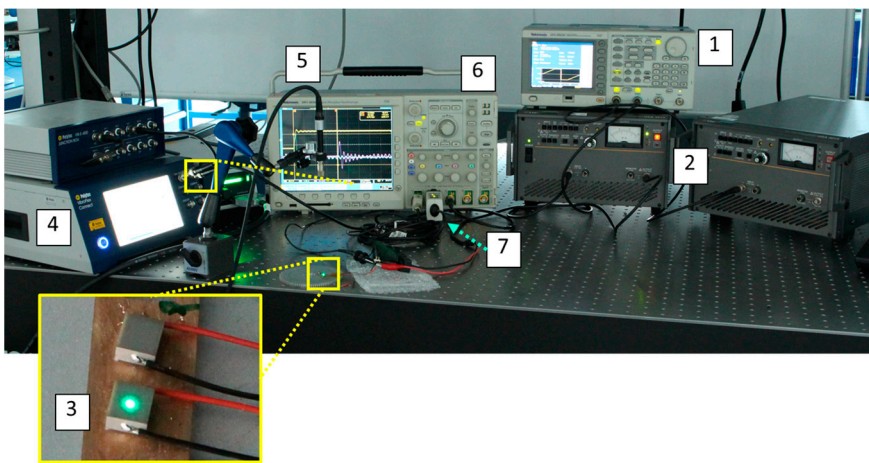

**Figure 13.** Measurement setup used to characterize the multilayer actuator (3) attached to a rigid mass. 1: Function generator, 2: Power amplifier, 3: Multilayer actuator ($3 \times 3 \times 2 \text{ mm}^3$, 4: Vibrometer controller, 5: Fiber-optic laser vibrometer sensor head, 6: Oscilloscope, 7: Current Probe.

An example of the raw data from these waveforms is shown in Figure 14 for a slip time of 1 µs. The collected data were then synchronized. Note that the transient current and the vibration velocity waveforms have the same shapes, including their overshoot peak values. The vibration velocity shown in Figure 14 has a gain factor of five. This value is nothing but the inverse of the force factor given in Equation (9). However, we should mention that this force factor value is valid only in transient cases. Whether continuous or at a steady state, driving a multilayer actuator at such a high vibration velocity would not be feasible. The time difference (approximately 40 µs) measured between the current and the vibration velocity peaks is due to the processing time of the laser vibrometer.

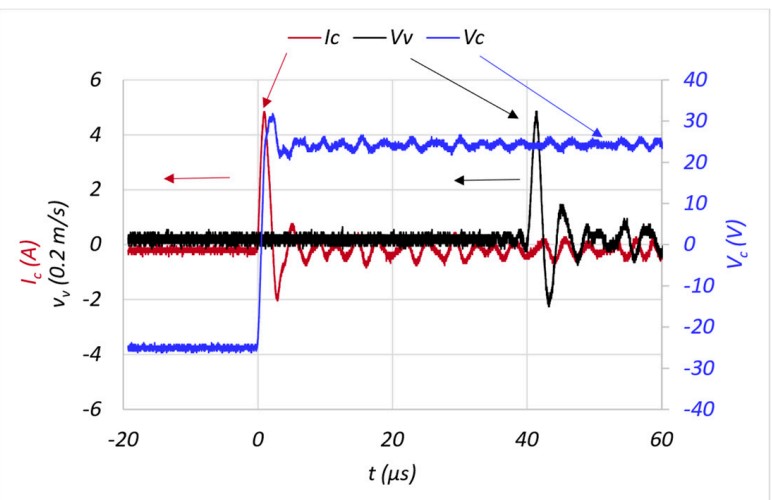

**Figure 14.** Measured raw data seen on the oscilloscope screen showing the transient current, vibration velocity and voltage waveforms. Vibration velocity values have a gain of five.

Figure 15 shows the measured transient voltage waveforms for different slip times. Even if the magnitudes of the voltages were set from −25 V to 25 V at the output of the power amplifier, due to the loading effect of the multilayer actuator, overshoots due to ramp inputs, especially for small slip times less than 3 μs, can clearly be seen.

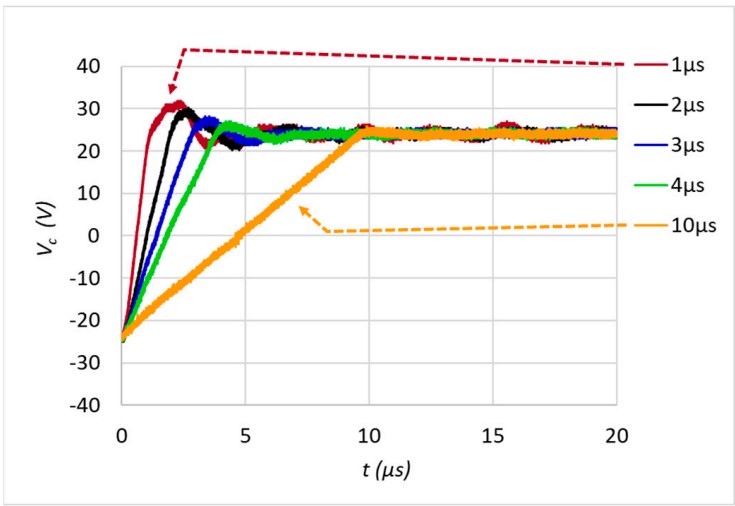

**Figure 15.** Measured transient voltage waveform of the multilayer actuator attached to a rigid mass.

A current probe (TCP0030A, Tektronix, Beaverton, OR, USA) was used to measure the transient currents on the multilayer actuator for different slip times. Figure 16 shows the transient current waveforms for slip times of 1, 2, 3, 4 and 10 μs. As with actuator voltage waveforms, the plot shows only a focused 20-μs-long section from the beginning of the slip time. Changes in the actuator currents for different slip times have the same tendency as the simulated values seen in Figure 10. The start times of the curves were set to the beginning of the slip time. While observing the current and voltage waveforms, we also measured the vibration velocity at the top of the multilayer actuator using the laser vibrometer (VibroFlex VIB-E-400, Polytec, Waldbronn, Germany). As shown in Figure 17, the transient vibration velocity curves have a similar progression to the measured current waveforms in Figure 16.

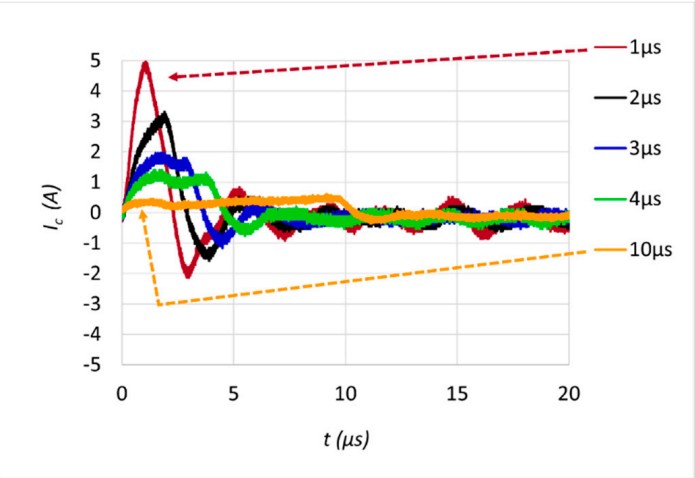

**Figure 16.** Measured transient current waveforms on the actuator.

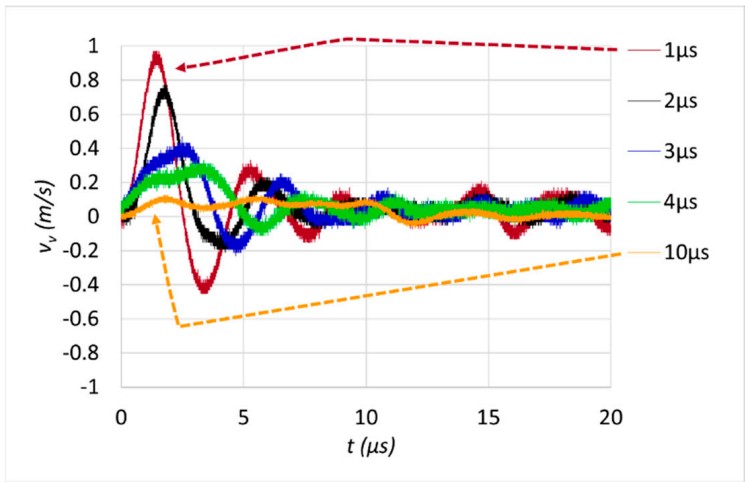

**Figure 17.** Measured transient vibration velocity waveforms at the top of the actuator.

*4.2. Transient Voltage and Current Waveforms during Slip Times (Stator in Motor)*

To characterize the transient voltage and current waveforms, we used a setup similar to that seen in Figure 13. First, a stator with six layers of pressing force springs (two in front and four at the back) was mounted onto a stage. This time, the function generator (Tektronix AFG3000, Beaverton, OR, USA) generated both anticyclic truncated sawtooth signals, and the signals were amplified by two power amplifiers (HAS 4014, NF Corp., Yokahoma, Japan). Single steps with different slip times were applied to the motor at 100 Hz. Voltage and current waveforms were observed, especially from the beginning of the slip times. The recorded voltage waveforms were later overlapped by setting the start of the slip times to 0.0 s. The measured voltage waveforms on the first and second actuators can be seen in Figure 18a,b). The total length of one period is 10,000 µs for the frequency of 100 Hz. These plots show only a focused section at the beginning of the slip time for 20 µs.

The current probe (TCP0030A, Tektronix, Beaverton, OR, USA) was used to measure transient currents for the different slip times of one multilayer actuator. Figure 19 shows the transient current waveforms applied to one actuator for slip times of 1, 2, 3, 4 and 10 µs. Similar to the voltage waveforms, the plot shows a focused section at the beginning of the slip time for 20 µs. Changes in the actuator currents for different slip times showed similar tendencies to the calculated values, as seen in Figure 10. Since the multilayer actuators in the motor are prestressed first in the stator and after mounting the stator in the motor, their peak current values are larger.

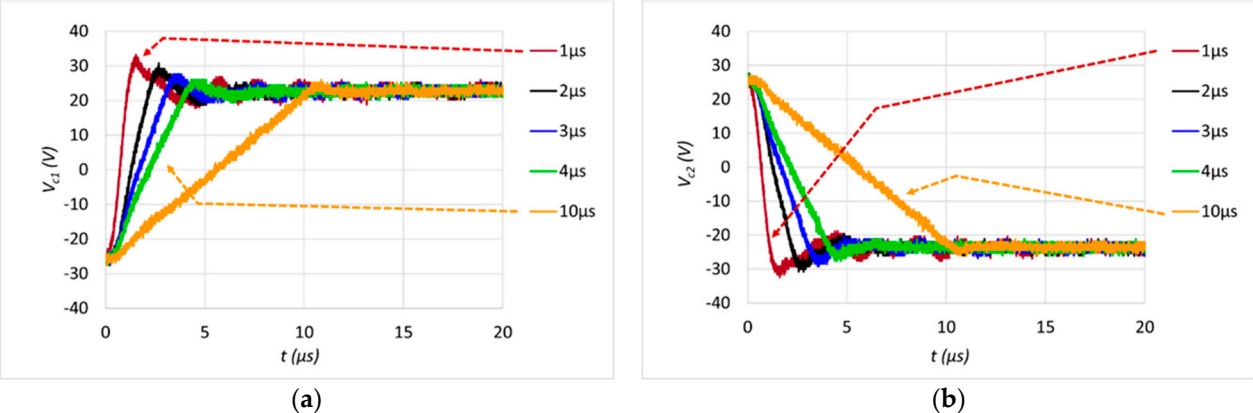

**Figure 18.** Measured transient voltage waveforms of actuator 1 (**a**) and actuator 2 (**b**).

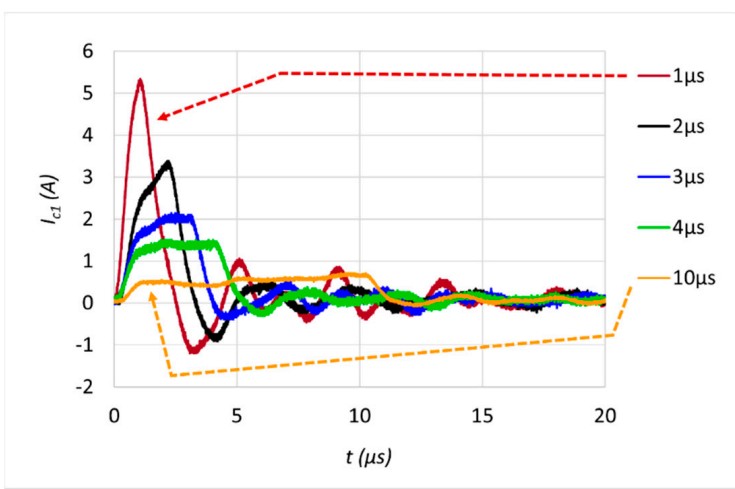

**Figure 19.** Measured transient current waveforms of one actuator in the stator.

## 5. Motor Characteristics

### 5.1. Load Characteristics

As mentioned previously, the stator is held on one side by the pressing force of the sheet metal flat springs, as seen in Figure 20a. When one side of the sheet metal flat spring layers is mounted on the motor base by two screws, a pressing force against the slider is generated (Figure 20b). Since the springs have relatively complex structures with straight and diagonal sections, the spring constant and thus the holding force of the slider with different numbers of springs could be obtained experimentally.

As epoxy materials are used to fix the multilayer actuators and friction couplers, the number of spring layers on the front side of the stator was constrained to two, and the number of spring layers on the back side was increased from two to six (in increments of one). Under these five conditions (2 + 2, 2 + 3, 2 + 4, 2 + 5, 2 + 6), the change in slider holding force depending on the different numbers of springs can be seen in Figure 21. When the stator was mounted on the motor base (Figure 20b) using two screws, the deformation of the sheet metal flat springs was 0.5 mm. Each additional spring increased the pressing force between the friction coupler and the slider by approximately 4.0 N, which raised the holding force by approximately 0.5 to 0.7 N. When there were two spring layers at the front and two at the back (2 + 2), the holding force was approximately 2.2 N. When an additional spring was mounted to the back (2 + 3), the holding force was increased to 2.8 N. Adding further layers at the back increased the force by approximately 0.7 N, as shown in Figure 21.

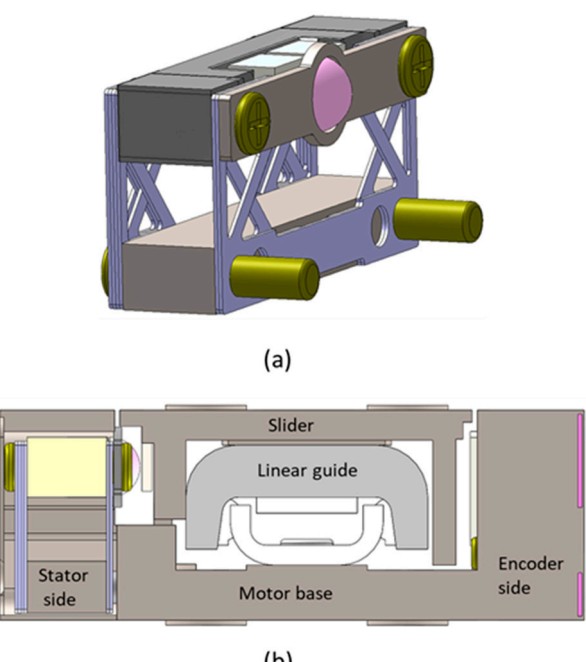

(a)

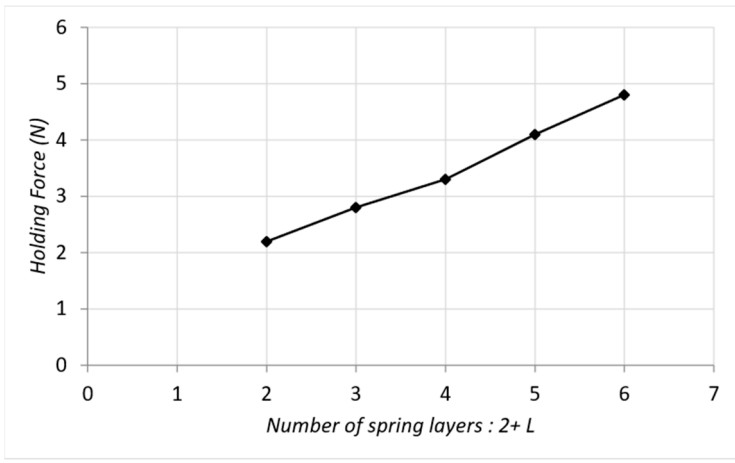

(b)

**Figure 20.** Perspective view of the stator (**a**) and end view of the inertia-drive-type motor (**b**). The stator has six spring layers, two at the front and four at the back (2 + 4). The pressing force springs on the front side were always constrained to two, and the number of spring layers at the back could be decreased or increased to change the pressing force, thus holding force of the slider.

**Figure 21.** The slider holding force was measured for different pressing force spring configurations; the number of spring layers (L) on the back of the stator was 2, 3, 4, 5 or 6.

To determine the load characteristics of the motor, truncated and anti-cyclic sawtooth signals at 20 kHz and +/−25 V were applied to the multilayer actuators in the stator. The function generator (Tektronix AFG3000 Arbitrary/Function Generators, USA) generated two synchronous anticyclic sawtooth signals with 1.0 µs of slip time. Additionally, two bipolar power amplifiers with bandwidths of 1.0 MHz (HSA 4011, NF Corp., Yokahoma, Japan) amplified the signals. The position sensor that is an optical incremental encoder with interpolation electronic board is already integrated into the motor at the other side of the linear guide captured the slider position data. Resolution of the encoder is 1 nm. While measuring the slider positions, the motors were operated under different loads. The slider velocities were obtained by computing the time derivative of the recorded slider positions. The effect of the pressing force springs on the load characteristics can be seen in Figure 22. The no-load velocity did not change with a higher pressing force exerted

by additional springs. However, the push-pull force capacity increased with the additional springs. To avoid wear on the interface of the friction coupler and the slider, we did not increase the pressing force further by inserting additional spring layers. Maximum thrust for (2 + 4) condition seems to be exceeding the holding force of 3.3 N. The reason would be due to the nonlinearities between the friction coupler against the slider contact.

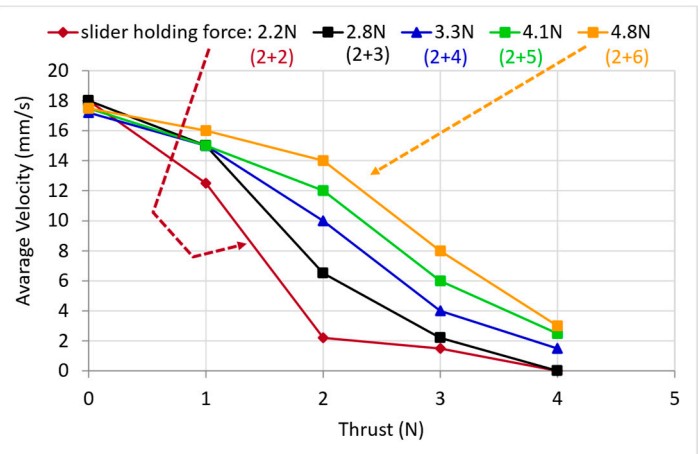

**Figure 22.** Load characteristics and effect of additional sheet metal flat springs at the back of the stator. Each additional spring increased the holding force by approximately 0.5 to 0.7 N.

By simply multiplying the average velocity with the pulling loads (or with thrust), it is possible to determine the instantaneous mechanical power (velocity × force), as seen in Figure 23. The instantaneous mechanical power curves are stabilized after the (2 + 4) spring configuration (two layers in the front and four at the back), which corresponds to a holding force of approximately 3.3 N. The instantaneous mechanical power measurement curves can be used to predict the closed-loop performance of an inertia-drive-type piezoelectric motor. Typically, the curves increase to a maximum power level and start to decrease as the force increases. The feasible range of stable closed-loop operation is approximately half of the maximum pulling force value.

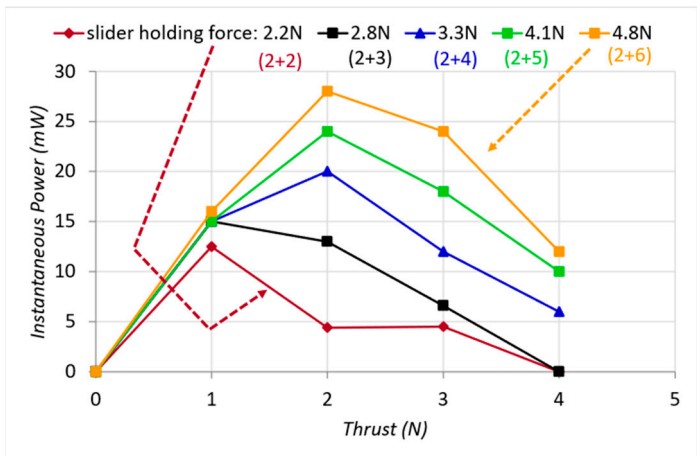

**Figure 23.** Instantaneous mechanical power output and the effect of additional sheet metal flat springs at the back of the stator.

### 5.2. Slider Transient Response during Slip Times

It is obvious that the friction coupler and the slider move together during the slow phase (stick time). Due to the rate of change of the current at the very beginning of the slip time, the acceleration and hence the force acting on the friction coupler reach a maximum.

When the force acting on the friction coupler exceeds the static friction force between the friction coupler and the slider, slippage occurs. Furthermore, we observed that the system response defines the amount of time that the acceleration acting on the friction coupler is required to reach a maximum value. Additionally, the magnitude of the force acting on the friction coupler is inversely proportional to the slip time. As seen in Equation (7), transient impact force on the friction coupler is proportional to the rate of change of the transient impact current. Since piezoelectric actuators are known for their fast response time (in the range of microseconds), at the very beginning of the slip time, slippage between the friction coupler and the slider occurs. For our stator geometry and the pressing force between the slider and the stator, return movements are eliminated for slip times of less than 3.0 μs. This behavior was also observed experimentally, as shown in Figure 24. Figure 24 shows the single step movements when the motor was operated at 100 Hz, the curves show the slider position from the beginning to the end of a single pulse. When the generated peak value of the force acting on the friction coupler did not exceed the static friction force, the slider moved together with the friction coupler. Due to the addition of the slider mass, the response time of the movement was much longer than the response time of the multilayer actuators.

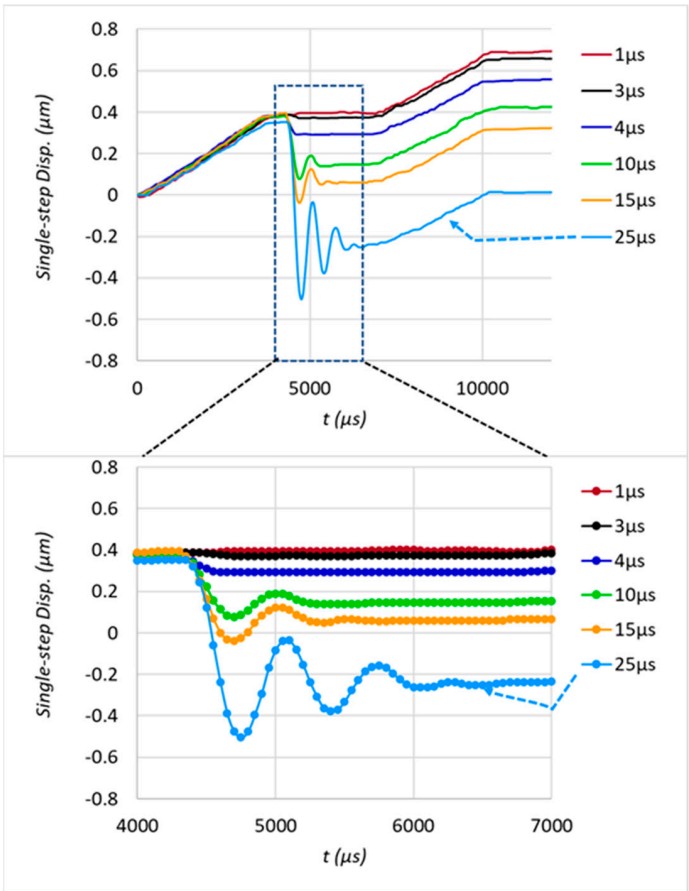

**Figure 24.** Measured single step slider positions and return movements of the slider for different slip times. When the slip time was greater than 3.0 μs, a return motion of the slider was observed. Even if the slip time is the dominant factor, a return motion can be holding force and operating frequency dependent. Because of the friction nonlinearities and state of the slider (on move or at rest) return movements amount can also vary. The tendency here is the longer the slip time the more the amount of the return movement.

Even if a longer slip time seems to be a disadvantage, slip time dependency on the slider return movement can be used to obtain high-precision small steps, provided that the drive electronics can generate signal waveforms with controllable slip times.

As shown in Figure 12, estimated peak value of the actuation force for 3 μs slip time for one multilayer actuator was approximately 1.6 N, which makes the total actuation force to be 3.2 N, which is approximately equal to the holding force of 3.3 N for (2 + 4) condition. Then having no slider return motion in Figure 24 for slip time of less than 3 μs is reasonable.

### 5.3. Mean Step Size at Different Frequencies

When designing the stator structure, an important criterion was to avoid the usage of any flexure or leverage so that the possible eigenmode frequencies of the stator were as high as possible. As shown in Figure 25, relatively constant mean step sizes of 0.8 μm were obtained from the 0.5 to 20 kHz range. To characterize the mean step size, the motor was operated for 100 steps in both directions, five times, during which the slider was not carrying any load. The mean step sizes were calculated from the movement amount at each frequency.

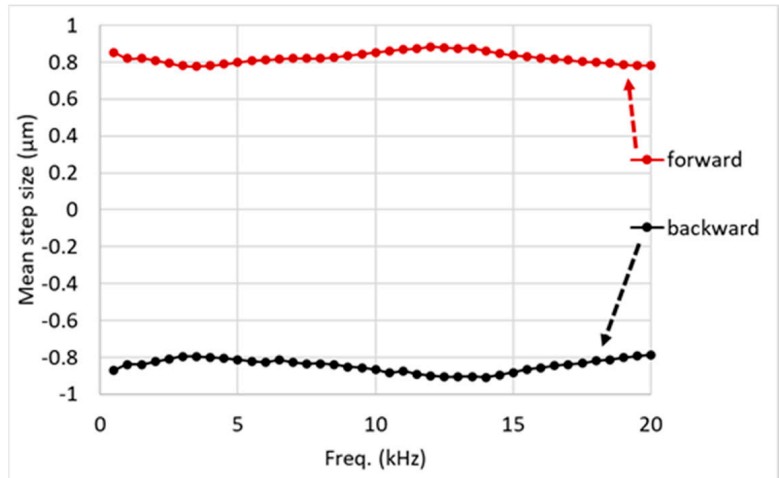

**Figure 25.** Open loop mean step size measurements when the motor was operated in forward and backward directions at frequencies ranging from 0.5 kHz to 20 kHz.

### 5.4. Open Loop Single Step Movements under Various Loads

For further characterization, the motor was operated with single step burst waveforms under various loads against gravity. As shown in Figure 26, movements in forward and backward directions are quite near for 0 N and steps are visible for a load of 3 N.

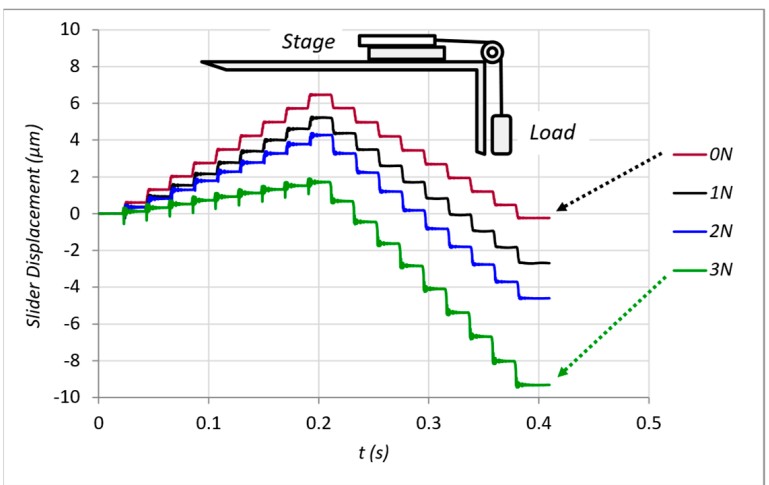

**Figure 26.** Open loop forward and backward step movements under various loads against gravity when the motor was excited with single-step burst driving waveforms at 20 kHz.

## 6. Discussion and Features

In this motor, one actuator in the stator always drives the slider in a forward mode, and the other in reverse mode. When a stator has been built properly and the multilayer piezoelectric actuators included in the design are sufficiently identical, it is possible to move the same amount in a forward or backward direction. Indeed, almost identical forward and backward steps can be achieved by adjusting the actuator potentials. In other words, assembly tolerances can be eliminated as necessary by a proper drive electronics. Moreover, by attaching the friction coupler toward one side of the middle position, a motor can intentionally be made stronger in one direction. Additionally, deformations on the multilayer actuators are converted into a rotational movement of the friction coupler directly without any flexure or leverage. This results the operating frequency of the motor to be as great as 30 kHz.

As the operating frequency of the motor can reach 30 kHz, the reliable operation of multilayer actuators at such high operating frequencies is an important issue. Our analysis revealed that the force acting on the friction coupler increased with reduced slip time; however, piezoelectric multilayer actuators are exposed to a transient vibration velocity as high as 1.0 m/s. This vibration velocity level makes the inertia-type motors dynamic but not quasi-static, as commonly accepted in the literature. To protect the multilayer actuators from delamination, the framed multilayer actuators in the stator are subjected to compressive force from the stator's pressing force springs and the top beam.

As one actuator expands and the other shrinks due to the anticyclic nature of the sawtooth signals, the hysteretic characteristics of the multilayer actuator movements cancel each other out, meaning that the movement of the friction coupler can be almost hysteresis-free [32].

Holding and pressing force springs are highly integrated into the structure of the stator. Mounting a stator on a motor base involves tightening two screws.

We found that, especially during slip time, the dynamic characteristics of the piezoelectric actuators play a critical role in obtaining high-quality steps without a return movement. Even if the force created by the actuators is conventionally believed to be directly dependent on the applied voltage, we observed that the force generated by the actuators during slip time is interrelated to the rate of change of the current absorbed by the actuators. Additionally, the slider return motion can be eliminated using shorter slip times.

When the power amplifier fulfills the required current absorbed by the multilayer actuators, then the force generated by the actuators can easily exceed the friction force and leave a slider behind during slip time. Therefore, the bandwidth of a suitable power amplifier must be defined by the slip time slope and not by the operating frequency of a sawtooth signal.

The current waveforms observed during slip time showed that the force acting on the friction coupler reached a maximum value. For very short slip times, the peak of the absorbed current and force acting on the friction coupler is large. During this very short period, the slider cannot follow the friction coupler when the slip time is shorter than 3.0 μs. As a result, minimal backward movement of the slider was recorded. However, by controlling the slip time, the return motion of the slider can be controlled. During the last sequence of the stick phase, very precise steps can be possible.

## 7. Conclusions

In this study, a new characterization method for multilayer actuators was proposed and a thickness-mode force factor expression ($A_{33}$) was introduced. The proposed thickness-mode force factor expression is defined in electrical domain. The proposed characterization method describes actuator behavior during slip time. Our analysis revealed that the impact force acting on the friction coupler produced by the actuators in an inertia-type piezoelectric motor is proportional to the rate of change in the transient current during slip-time.

We also reported that a vibration velocity on a multilayer actuator can reach up to 1 m/s during slip-times. This observation can initiate the usage of multilayer actuators made with hart type piezoelectric materials in inertia type piezoelectric motors.

**Author Contributions:** Conceptualization, B.K. and B.D.; methodology, B.K.; validation, B.K. and B.D.; formal analysis, B.K.; investigation, B.K.; resources, B.K.; data curation, B.K.; writing—original draft preparation, B.K.; writing—review and editing, B.K. and B.D. All authors have read and agreed to the published version of the manuscript.

**Funding:** This research received no external funding.

**Institutional Review Board Statement:** Not applicable.

**Informed Consent Statement:** Not applicable.

**Data Availability Statement:** Not applicable.

**Conflicts of Interest:** The authors declare no conflict of interest.

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
