# Peer review of "Impact Force Analysis in Inertia-Type Piezoelectric Motors"

_actuators, doi:10.3390/act12020052_

Round 1

Reviewer 1 Report

This paper reports on the research and development of an inertia type piezoelectric motor driven by the multilayer actuators. For performance evaluation of the motor, numerical and experimental investigations of linear piezoelectric motor based on two multilayer actuators connected by arc shaped coupler are performed. In this study an actuator’s characteristics were investigated while asymmetric saw tooth driving signals are used. The proposed piezoelectric motor is analyzed theoretically as well as experimentally and it will contribute to progress in applications of the precision positioning systems.

Overall, I find the paper to be useful and well-executed. There are a few items that need attention. Specific comments and questions needing attention are provided below.

Specific Comments:

1. Length of abstract should be reduced with goal briefly represent topic of the manuscript as well as obtained results.

2. An author’s states “…the motor consists of two identical multilayer actuators.” Please, describe in detail, how two identical multilayer actuators were obtained/manufactured i.e., geometry, capacitance, damping etc.

3. Does square shaped signal (PWM) with differential width and controlled fall and rise times can be used to drive the actuator? If so, why an author’s decided to use more complex, from viewpoint of electronics, signal shape?

4. How does reaction force from slider/load effects operation of the motor? Especially in stepping /fine motion modes.

5. Page 6, line 222: a sentence Since multilayer actuators are a…” should be corrected.

6. Please explain the meaning of frequency of the driving signal presented in Fig.4, Fig.5 and Fig.26. In Fig. 5 period T or timespan of the composite signal is 10 ms and frequency of stepping motion is 20Hz (period – 50 ms). In Fig. 26 a frequency of stepping motion is 50Hz (period – 20 ms).

7. Page 17, line 499: more information about the position sensor integrated into the motor should be presented.

8. Numbering of Figure 20 is improper.

9. An author’s states: Line 614 “…almost hysteresis free” The manuscript does not contain modeling or experimental research on motion/deformation hysteresis. This statement is assumption which is not directly covered by any background. Moreover, the springs/plates which are used to generated preload force is direct source of possible hysteresis especially at higher preload forces or deformation levels.

Author Response

Please find our response to the suggestions and comments of the reviewers in “Response_to_Reviwers.pdf” file.

Reviewer 2 Report

This paper was well-organized and written. However the reviwer has been found some mistakes in the manuscript. Please modify as follows:

1. The schematic dimensions of the stator, slider, hemisphere and piezo stack.

2. Don't indent 'where' in the next line of the fomulas.

3. In Eq. (6), the mass m_p is the equivalent mass of the piezo element, because the one end of the piezo element is fixed. The equivalent mass is 1/3 of the mass of the piezo element.

4. In 3.2 Eauivalent Circuitry, explain how to obtain the values of R, C and L.

5. Fig. 8 is not mentioned in the text.

6. Part of Fig. 18 has been erased. Modify it.

7. Figure 20 is misprinted as Figure 1

Author Response

Please find our response to the suggestions and comments of the reviewers in the “pdf” file.

Reviewer 3 Report

The authors of the paper entitled "Impact Force Analysis in Inertia-Type Piezoelectric Motors" presents an investigation into an inertial piezoelectric motor model developed in-house. The work is very well elaborated from the perspective of the scientific content. Several objections that can contribute to an increase in the quality of this work:

  1. It must be reduced abstractly to no more than 200 words according to the general rules, a large part of the current content must be transferred to the INRODUCTION section.
  1. You must respect the spacing according to template by using the Styles facility in Word.
  1.  For Figures ditto
  1. Figure 9 and 10 can be converted into Figure 9(a), (b)
  1. Ditto for Figure 11, 12
  1. Figure 13 must be restored, without the list of equipment included in the figure. These will be listed based on the references in the text associated with the Figure.
  1. Figures 15 and 16 can be converted into Figure x (a), (b)
  1. Row: 508 The figure is wrongly numbered must be restored.
  1. The work has no conclusions, they are mandatory.

A neater elaboration of the work by eliminating the elements listed above will offer readers a more compact form and easier to follow by the readers of the journal. The scientific content of the article is of quality results being obtained in an industrial environment of professional reference.

Author Response

(The authors gave the same response as above.)

Reviewer 4 Report

In this work, a two-phase inertia-drive-type motor consists of two commercially available multilayer actuators is introduced. The topic is interesting and meaningful. However, some points should be improved. 1. The abastract is too long, please follow the requirements of the journal. 2. More related refences should be given, especially recent years. 3. For the driving signal, why P3 is necessary? For the sawtooth signal, usually only P1 and P2 is enough.
4. How about the influence of the resonant frequency to this motor?
5. The data figures should be improved, they are not good arranged.

Author Response

(The authors gave the same response as above.)

Round 2

Reviewer 3 Report

The authors of the paper entitled " Impact Force Analysis in Inertia-Type Piezoelectric Motorsin the form revised in accordance with the previous recommendations are presented in an improved form. The authors' comments on previous observations provide a clearer picture of the contributions contained in this paper. In its current, revised form, the paper has the necessary qualities to be recommended for publication, as it is.